# On the Unfounded Enthusiasm for Soft Selective Sweeps III: The Supervised Machine Learning Algorithm That Isn’t

**DOI:** 10.3390/genes12040527

**Published:** 2021-04-05

**Authors:** Eran Elhaik, Dan Graur

**Affiliations:** 1Department of Biology, Lund University, Sölvegatan 35, 22362 Lund, Sweden; 2Department of Biology & Biochemistry, University of Houston, Science & Research Building 2, Suite #342, 3455 Cullen Bldv., Houston, TX 77204-5001, USA; dgraur@uh.edu

**Keywords:** artificial intelligence (AI), supervised machine learning (SML), evolutionary biology, molecular and genome evolution, selective sweeps, population size

## Abstract

In the last 15 years or so, soft selective sweep mechanisms have been catapulted from a curiosity of little evolutionary importance to a ubiquitous mechanism claimed to explain most adaptive evolution and, in some cases, most evolution. This transformation was aided by a series of articles by Daniel Schrider and Andrew Kern. Within this series, a paper entitled “Soft sweeps are the dominant mode of adaptation in the human genome” (Schrider and Kern, *Mol. Biol. Evolut*. **2017**, *34*(8), 1863–1877) attracted a great deal of attention, in particular in conjunction with another paper (Kern and Hahn, *Mol. Biol. Evolut*. **2018**, *35*(6), 1366–1371), for purporting to discredit the Neutral Theory of Molecular Evolution (Kimura 1968). Here, we address an alleged novelty in Schrider and Kern’s paper, i.e., the claim that their study involved an artificial intelligence technique called supervised machine learning (SML). SML is predicated upon the existence of a training dataset in which the correspondence between the input and output is known empirically to be true. Curiously, Schrider and Kern did not possess a training dataset of genomic segments known *a priori* to have evolved either neutrally or through soft or hard selective sweeps. Thus, their claim of using SML is thoroughly and utterly misleading. In the absence of legitimate training datasets, Schrider and Kern used: (1) simulations that employ many manipulatable variables and (2) a system of data cherry-picking rivaling the worst excesses in the literature. These two factors, in addition to the lack of negative controls and the irreproducibility of their results due to incomplete methodological detail, lead us to conclude that all evolutionary inferences derived from so-called SML algorithms (e.g., S/HIC) should be taken with a huge shovel of salt.


*“So, off went the Emperor in procession under his splendid canopy. Everyone in the streets said, ‘Oh, how fine are the Emperor’s new clothes! Don’t they fit him to perfection? And see his long train!’ Nobody would confess that he couldn’t see anything, for that would mean that they are fools. No costume the Emperor had worn before was such a complete success.*
Hans *Christian* Andersen’s *Keiserens Nye Klæder* (1837), translated by Jean Hersholt

## 1. Introduction

For most of the history of the population genetics discipline, soft selective sweeps were considered an obscure evolutionary mechanism of little importance in species with large effective population sizes and of no importance at all in species with small effective population sizes, such as human populations. In the last 15 years, however, soft selective sweeps have metamorphosed from a curiosity into a ubiquitous mechanism that has been claimed to explain most adaptive evolution and, in some cases, most evolution in all life forms from viruses to humans (reviewed in Jensen [1] and Harris et al. [2]). 

A series of articles by Daniel Schrider and Andrew Kern [3,4,5,6] were particularly important in this metamorphosis. Within this series, a paper entitled “Soft sweeps are the dominant mode of adaptation in the human genome” [4] attracted a great deal of scientific and popular attention, in particular in conjunction with another paper [7] which made the headline-grabbing claim that the Neutral Theory of Molecular Evolution [8] needs to be rejected because it was based on “unreliable theoretical and empirical evidence from the beginning.” Both Schrider and Kern [4] and Kern and Hahn [7] were thoroughly refuted by Harris et al. [2] and Jensen et al. [1], respectively.

In this paper, we address a purported novelty in Schrider and Kern’s [4] paper, i.e., the claim that their study involved an artificial intelligence technique called Supervised Machine Learning (SML). In addition, our examination of this paper led us to the uncovering of additional questionable practices that were either not mentioned in Harris et al. [2] or were only mentioned briefly 

For purposes of clarity, before further dissecting Schrider and Kern [4], we outline the principles of machine learning and some features of selective sweeps.

## 2. Principles and Limitations of Machine Learning

The field of machine learning is concerned with the development and application of computer algorithms that improve with experience [9]. Machine learning methods are roughly classified into two primary categories: supervised and unsupervised machine learning. The main task of both machine learning categories is to predict the assignment of an input object (usually a vector of character states) to a label, where labels are the targets of the prediction task. 

Unsupervised machine learning methods are designed to identify patterns in the data without training, while supervised machine learning methods, which include support vector machines (SVMs), require a training dataset. Unsupervised machine learning algorithms require only unlabeled input data and the desired number of different labels to assign as input. For example, we may segment the human genome into equal-size segments of length 100,000 nucleotides, collect data on each segment, such as nucleotide composition, mean length of mononucleotide runs, and distance from the centromere, and ask the algorithm to assign each segment to one of five labels or one of 216 labels. We note that none of the labels in unsupervised machine learning can be named *a priori*, only *a posteriori*.

Supervised machine learning (SML) algorithms aim to solve two types of problem: (1) the classification problem, whose target is a qualitative variable, e.g., determining whether a gene is expressed or not expressed under certain conditions, and (2) the regression problem, whose target is to predict a numerical value, e.g., the level of gene expression of a gene under certain conditions. In the following, we will only deal with the classification problem. 

Classification SML algorithms are predicated on the existence of a large set of labeled examples. SML algorithms are trained on labeled examples and are then used to make predictions on unlabeled examples. The set of labeled examples is called the “training dataset”. Classification SML algorithms use what they have learned from the training dataset to make predictions or decisions on a “test dataset”, i.e., a set of unlabeled examples. For example, if we wish to determine whether or not a gene is expressed, we need a “training dataset” of genes for which we have solid empirical evidence on their expression status. We can then unleash the machine learning algorithm on a collection of genes for which expression data is lacking.

The immense success that SML techniques have had in the world of e-commerce [10], as well as in some areas of medicine [11,12], genetics [13], genomics [14], and biochemistry [15] may have prompted Schrider and Kern to develop so-called supervised machine learning methodologies [3] to address evolutionary questions, particularly those aimed to clarify the relative importance of selection and random genetic drift during the evolution of genomes [4]. Such studies have been ballyhooed as “sophisticated”, “cutting-edge”, “robust”, and “valuable”, and it has been argued that they “make a strong case for the idea that machine learning methods could be useful for addressing diverse questions in molecular evolution” [16]. The method has since been applied to various other organisms besides humans (e.g., [17,18,19]).

Which type of machine learning should we use? In their influential review of machine learning application in genetics and genomics, Libbrecht and Noble [10] put forward some clear and simple rules on the use of supervised and unsupervised learning. In particular, they note that when a labeled training set is not available, one can only perform unsupervised learning. Interestingly, however, when labels are available, it is not always the case that the supervised machine learning approach is a good idea. This is because every supervised learning method rests upon the assumption that the distribution responsible for generating the training dataset is the same as the distribution responsible for generating the test dataset. This assumption is only met if a single labeled dataset is randomly subdivided into a training set and a testing set. 

SML works well when the training data are sufficiently detailed and comprehensive to represent the truth. When the data deviate from this optimum, so will the SML results. When the training dataset does not exist, as is the case in evolutionary studies where the past is unknown, it is impossible to use SML [10], and unsupervised methods should be preferred, though with extreme caution.

## 3. What Are Hard and Soft Selective Sweeps?

A hard selective sweep is a process in which a single novel beneficial mutation arises in a population and rises in frequency quickly to fixation. Patterns expected under the hard sweep model mainly include a reduction in variation surrounding the beneficial mutation owing to the fixation of the haplotype in which the beneficial mutation cropped up by chance and well-described effects on the allele frequency spectrum and patterns of linkage disequilibrium. 

By contrast, a soft selective sweep does not reference a particular population genetic model but rather a collection of many models that may result in similar genomic patterns of variation. Two such models that are banded together within the vague definition of “soft selective sweep” are selection on a preexisting variant in the population that is associated with two or more haplotypes (also referred to as “selection on standing variation”) and selection on multiple new mutations. As far as selection on standing variation is concerned, a selective shift is required, i.e., a variant that was previously neutral or deleterious becomes advantageous. 

Orr and Betancourt [20] and Hermisson and Pennings [21] defined the conditions under which a soft sweep from standing variation becomes possible. They found out that a soft selective sweep is feasible only if the beneficial mutation segregated at appreciable frequencies on the population before the selective shift either through balancing selection (if it was deleterious) or random genetic drift (if it was neutral). Interestingly, even if selection is acting on standing variation, in the vast majority of cases, a single haplotype will fix in the population, rendering what should have been a soft selective sweep indistinguishable from a hard selective sweep. Thus, on theoretical grounds, soft sweeps should be even rarer than hard sweeps and are unlikely to be observed, especially in species with small effective population sizes, such as humans. 

## 4. An Additional Look at the Methodology and Claims of Schrider and Kern (2017)

In studying the role of adaptation in recent human evolution, Schrider and Kern [4] claimed to have applied an SML approach to simulated human population data to evaluate the evidence for positive selection in humans. The authors used genomic data from six human populations and a so-called supervised machine learning procedure to arbitrarily classify large segments of the human genome (i.e., genetic windows) into five classes if they: (1) have experienced a hard selective sweep, (2) are linked to a hard selective sweep, (3) have experienced a soft selective sweep, (4) are linked to a soft selective sweep, or (5) have evolved neutrally, which includes regions evolving under strict neutrality as well as sequences affected by purifying selection. Their conclusion, as written in the title, was “Soft Sweeps Are the Dominant Mode of Adaptation in the Human Genome”. In the paper itself, the results are presented in a more nuanced manner; while Schrider and Kern [4] only found a minority of genomic windows to be subject to either soft or hard selective sweeps (7.6%), a large fraction of windows was found to be “linked to a completed selected sweep, either hard or soft (56.4% on an average).” Neutral evolution accounted for the evolution of only 36.0% of genomic segments. 

Harris et al. [2] examined the assumptions and the performance of Schrider and Kern’s [3] S/HIC algorithm used in Schrider and Kern [4] and concluded that the claims regarding the algorithm’s robustness and accuracy rely on prior knowledge of both the distribution of fitness effects as well as the demographic history of the population in question, but as neither is known in practice, rampant misclassification occurs. Thus, hard sweeps of weakly beneficial mutations, as well as neutral demographic models (e.g., bottlenecks, structured populations, and migration), are frequently misclassified as soft sweeps, while severe population bottlenecks tend to result in genomic patterns of variation that mimic hard sweeps. Furthermore, if the true demographic model is unknown, S/HIC tends to misclassify the vast majority of hard selective sweeps as soft sweeps.

## 5. The Supervised Machine Learning Algorithm That Isn’t

The simplest and most important reason for rejecting Schrider and Kern [4] is that their supervised machine learning simply isn’t. Schrider and Kern [4] possess no training dataset of genomic segments known *a priori* to have evolved either neutrally or through soft or hard selective sweeps. Nor can such a dataset ever exist. In the absence of the most crucial ingredient of supervised machine learning, Schrider and Kern [4] should be treated as a pretend study using pretend SML. Their so-called training dataset consists of simulated data based on input parameters inferred or adopted from the literature, which they treat as true labeled data. 

One might, of course, ask what is wrong with simulations? The answer is that there is nothing intrinsically wrong with simulations; they are the bread-and-butter of population genetic research since Francis Galton invented the quincunx or bean machine in 1894. Simulations, however, cannot be used in an SML context because they do not represent data that are known empirically to be true. In other words, simulation results are not true labels. The correct way to use simulated data is to present all the assumptions of the simulation and compare them to the empirical findings. Because of the asymmetry of statistical testing, however, if the empirical values differ from the simulated results, then one or more assumptions used in the simulation can be rejected as unrealistic or untrue; if, on the other hand, the empirical results and the simulated results are identical, the model used in the simulation—especially if it is a complicated model with many variables—can not be said to be true.

But, let us assume for a moment (and only for a moment) that simulated data can be used in SML. The appropriate method to test the validity of the results will be to use unsupervised machine learning with five unnamed labels and test whether any of these five unnamed labels coincide with Schrider and Kern’s five genomic classes.

## 6. Other Questionable Practices

The “mortal sin” of using a supervised machine learning algorithm with no training dataset should be enough to invalidate Schrider and Kern’s results and conclusion. However, on top of this *delictum*, Schrider and Kern piled a large number of sampling transgressions, logical missteps, internal inconsistencies, and corruption of statistical practices, which we present henceforth.

**Cherry-Picking Population Data.** To characterize selection in global human populations, Schrider and Kern [4] had first to choose which populations to use. Because populations with a large variation in demographic and historical parameters due to processes such as extreme bottlenecks, a history of migrations, and complex admixture or gene flow patterns would introduce substantial uncertainties in their subsequent simulations, they decided to exclude such populations from their sample. But have they?

Schrider and Kern [4] claimed to have selected six populations that experienced the minimal influence of admixture or migration: They included Gambians in Western Divisions in The Gambia (GWD) and Yoruba in Ibadan, Nigeria (YRI) from West Africa; Luhya in Webuye, Kenya (LWK) from East Africa; Japanese in Tokyo, Japan (JPT) from Asia; Utah residents with Northern and Western European Ancestry (CEU) from Europe; and Peruvians from Lima, Peru (PEL) from the Americas. They motivated their choice by the homogeneity of populations over various numbers of splits (*K*) in the ADMIXTURE analysis [22] carried out by Auton et al. ([23], Extended Figure 5): *“We see that for most values of K, each of these populations appears to correspond primarily to a single ancestral population rather than displaying multiple clusters of ancestry”.*


An examination of the ADMIXTURE results of Auton et al. shows that Schrider and Kern [4] frequently violated their own selection criteria. For example, CEU, a population of unclear origins that shows multiple splits over various *K* values, was included, while the more homogeneous British (GBR) and Finnish (FIN) populations were excluded from their analysis. All the three African populations included in the study yield larger and more numerous splits compared to the Esans (ESN), the most homogeneous African population, which was excluded from their study. Finally, the Chinese (CDX) population was excluded despite its genetic homogeneity.

Schrider and Kern [4] admitted to one exception by including the cross-continentally admixed PEL population because “among the highly admixed American samples it appears to exhibit the smallest amount of possible mixed ancestry (for most values of *K*)”, so they “retained this population to have some representation from the Americas”. Oddly, this “exception” clause was not applied to South Asian populations, which represent a considerable fraction of human global genetic diversity. 

The choice of populations is further questionable on several grounds. First, Schrider and Kern’s [4] inference of populations with “single ancestral population rather than displaying multiple clusters of ancestry” is doubtful since ADMIXTURE is unreliable when used to determine whether groups are pure representatives of one ancestral source and was certainly not designed to test whether two populations have mixed. Even groups that appear to be extremely homogeneous are known to have contributions from ancestrally related groups [24]. Second, applying ADMIXTURE analysis with an arbitrary number of splits yields inconsistent splits across the population panel. The appearance of “heterogeneity” is driven by populations with distinct combinations of allele frequencies rather than true homogeneity. Finally, considering the inconsistency between the selection criteria outlined by Schrider and Kern [4] and their sample population set, it is likely that these criteria were applied *post hoc* to match populations already included in the dbPSHP database [25], which the authors later employed to study selection. dbPSHP has no data for the most homogeneous populations in Auton et al. (GBR, FIN, ESN, and CDX) but has data for four of the six populations analyzed by Schrider and Kern [4]: YRI, LWK, CEU, and JPT. Excepting Gujarati Indians in Houston, Texas (GIH), dbPSHP excludes all South Asians. 

To summarize, Schrider and Kern [4] did not follow their own criteria for selecting populations with the least complex demographic histories. Instead, Schrider and Kern [4] employed a type of observational bias, called the “streetlight effect” [26], by only searching for preconceived answers where it is easiest to find them (i.e., in populations for which selection data were readily available).

**Estimating the Parameters of the Demographic Model**. To detect sweeps, Schrider and Kern [4] applied a maximum likelihood approach that considers simulated genomic patterns using a variety of population genetic summary statistics and classifies genomic windows as being: (1) the target of a completed hard sweep (hard), (2) closely linked to a hard sweep (hard-linked), (3) a completed soft sweep (soft), (4) closely linked to a soft sweep (soft-linked), or (5) to have evolved neutrally (neutral). Since no training dataset was available for exploratory evolutionary studies, Schrider and Kern [4] developed a demographic model that disgorged a number of summary statistics for genomic regions that have experienced simulated hard sweeps, simulated soft sweeps, or have evolved under simulated neutrality. 

We emphasize again that unlike in normal SML approaches, where training and testing of the classifier are two separate operations, in Schrider and Kern [4] the classifier was never trained. Notwithstanding this glaring insufficiency, the authors use the terms “training” eleven times in their article. For instance, they claim that “training examples for the hard class experienced a hard sweep in the center of the central sub-window” despite the fact that the “training” data were not actual data as required for SML applications, but rather simulated “data” created in, as we will demonstrate, an extremely problematic manner. The use of terms such as “machine learning” and “classifier” is, thus, entirely inappropriate. In the next section, we show that even as far as building their faux “classifier” and estimating the variables on which it was based, the authors made some very questionable decisions. 

**Estimating Demographic History from Genomic Data**. Before selection can be simulated on the genomic data, Schrider and Kern [4] had to reliably capture population-size changes over discrete time intervals for their coalescent simulation tool *discoal* to work properly [27]. For that, they turned to the demographic model calculated by Auton et al. [23] using the Pairwise Sequentially Markovian Coalescent (PSMC) model [28]. PSMC employs coalescent methodology to reconstruct changes in the effective population-size history over time under neutrality. Schrider and Kern extracted 26 discrete points per population from Auton et al.’s extended Figure 5 (Figure 1A), scaled them by the mean population mutation rate *θ* of the population under study (which they determined, as we show in the next section) and by the present-day effective population size *N_0_* (10,000), and included them in their simulation (the -*en* parameter) (Figure 1B). This simple procedure, which should have resulted in a similar demographic model to the one used by Auton et al., has instead resulted in inflated population sizes by a factor of up to 10^4^ and a complete distortion of Auton et al.’s demographic model. We note that PSMC’s output cannot always be reliably interpreted as plots of population-size changes, particularly if the population is admixed, in which case peaks on the demographic plot might correspond to periods of increased population structure rather than increased population size. Remarkably, Schrider and Kern [4] noted that “these models may not accurately capture the demographic histories of the populations we examined.” However, they claimed that their Soft/Hard Inference through Classification or S/HIC methods [27] is robust to “demographic misspecification”, and hence did not expect this factor “to severely impact” their analysis. 

**Estimating The Population Mutation Rate (*θ*).** The population mutation rate (*θ*) is a fundamental parameter in evolutionary biology as it measures the average genomic mutation rate of the entire population or, stated differently, it describes the amount of selectively neutral diversity in a population [29]. It is also necessary to estimate the effective population size (*N_e_*). *θ* is calculated as 2*pN*_e_*μ*_tot_, where *p* = 1 or 2 for haploids and diploids, respectively, *N*_e_ is the effective population size, and *μ*_tot_ is the mutation rate at the locus of interest. Schrider and Kern [4] assumed that the present-day grid of *θ* ranged from 10 to 250 and was calculated from 4*N*_e_*μL*. Here, *μ* is the mutation rate per nucleotide, which Schrider and Kern [4] set as 1.2 × 10^−8^ per base pair per generation as was calculated by Kong et al. [30] (based on Icelandic trios for all their populations), and *L* is the length of the segment.

Confusingly, multiple *L* values were used: 100,000, the only one reported in the paper; 200,000, which was used in the code (Table S5 in [4]); 2,200,000, which was used for their Table S1 (D. R. Schrider, personal communication); and 1,100,000, the “correct” value for this analysis which was never reported (D. R. Schrider, personal communication). In their paper, Schrider and Kern [4] neither disclosed the range of *L* values nor explained them. Using the latter value of *L*, *N_e_* = 1500 for Africans and *N_e_* = 4250 for non-Africans from [23], *θ* should have a theoretical range of 79 and 224, similarly to the proposed grid. Unfortunately, neither these nor the stated grid values of *θ* (10–250) were used. In practice, Schrider and Kern [4] employed *θ* values that ranged from 40 to 2200 (their Table S5, *-Pt* parameter) since they chose “as the final values of *θ* that for which the sum of the percent deviations of the simulated from the observed means of each statistic was minimized.” By inflating the range of *θ*, Schrider and Kern [4] have artificially modified the genomic mutation rate providing more opportunities for “selection” to act. As we shall next show, this choice also had the effect of bloating the effective population size.

**Estimating the Effective Population Size (*N_e_*).** Deriving *N_e_* from the two extreme values of *θ* (40 and 2200) and the published *L* value (*L* = 100,000) yields a remarkable *N_e_* range of 8333 < *N_e_* < 458,333, i.e., values similar to those of *Plasmodium falciparum* (210,000 < *N_e_* < 300,000) and *Drosophila melanogaster* (*N_e_* = 1,150,000) [31]. Even using the “correct” *L* value (*L* = 1,100,000) (D. R. Schrider, personal communication) results in an extreme range for *N_e_* 757 < *N_e_* < 41,666, compared to the *N_e_* calculated by Auton et al. (1500 < *N_e_* < 4250) [23] (Figure 1A), which Schrider and Kern (2017) supposedly relied on for their demographic model and the well-established range for humans (1000 < *N_e_* < 10,000) (e.g., [32,33,34,35]). This *N_e_* is also higher than in most apes (7 < *N_e_* < 20,000), excepting Central chimpanzee (25,000 < *N_e_* < 100,000) [36]. By inflating *N_e_*, Schrider and Kern [4] have biased all their subsequent calculations.

**Estimation of the Population Recombination Rate (*ρ*).** To understand the magnitude of the bias in Schrider and Kern’s [4] analyses, let us consider their simulation of the population recombination rate, *ρ* = 4*N_e_r*, where *r* is the crossover rate per base pair. The distribution of *ρ* was empirically found to be “unambiguously unimodal”, with a mean of 10^−4^ ([37], Figure 3). Further, *ρ* is normally distributed with 80% of its values in the range of 10^−4.5^ to 10^−3.5^ ([37], Figure 3). In contradistinction, Schrider and Kern [4] supposedly derived *ρ* from an exponential distribution with a mean of 1 × 10^−8^, although their Table S5 (-*Pre* parameter) indicates, again, that different values of *ρ* were used (Table S5, -*Pre* parameter). According to that table, the mean of the exponential distribution ranged from 183 to 1008. 

**Simulating Genomic Sequences.** Deciding on the demographic model parameters for each population, Schrider and Kern [4] have next generated genomic sequences using several approaches that we have attempted to replicate. Replication is the “cornerstone of science” and its most fundamental principle; the inability to replicate published studies has been termed the “replicability crisis” [38]. We found that Schrider and Kern’s [4] study is part of this crisis. 

The Methods section in Schrider and Kern [4] renders replication impossible. First, the authors employed various computational tools, only some of which are mentioned in the paper. Second, they provided the code for only one of the tools (Table S5 in [4]). And finally, their description of the simulation is partial at times, erroneous at other times, and inconsistent with the code they have provided. 

To name a few examples, Schrider and Kern [4] wrote that “we used the program *discoal* [27] to simulate large chromosomal regions, subdivided into 11 sub-windows”. However, *discoal* simulates very short genomic regions (in a setting of 200,000 bp, *discoal* simulated regions varied in size from 0–2600 bp with a mean of 934 bp) and the subdivision into windows is part of a different package, called S/HIC (which, however, is no longer available where it was supposed to be deposited, https://github.com/kern-lab/, accessed on 31 March 2021). To understand whether these numbers are reasonable, we downloaded the CEU data from the 1000 Genomes Phase 3 (v5) [23]. We sampled 200,000 regions from all autosomal chromosomes 1000 times and counted the number of SNPs. The number of SNPs ranged from 176 to 6277, with a mean of 2195 and a median of 2134. It was never 0, as in the *discoal* simulations. In other words, the number of SNPs in the simulated *discoal* sequences is less than half than in reality.

In another place in the article, the authors wrote, *“we simulated additional test sets of 1000 genomic windows 1.1 Mb in length with varying arrangements of selected sites”,* without telling the reader which simulation tool they used. 

Another typical paragraph reads: *“The simulation program discoal requires some of these parameters to be scaled by the present-day effective population size; we did this by taking the mean value of θ and dividing by 4 uL, where u was set to 1.2 × 10^−8^ [30]. The full command lines we used to generate 1.1 Mb regions (to be subdivided into 11 windows each 100 kb in length) for each population are shown in Supplementary Table S5, Supplementary Material online. We also simulated 1000 test examples for each population in the same manner as for the training data”.* This section describes the use of at least three tools with only discoal referenced explicitly. It remains unclear how many regions were simulated since parts of the manuscript mention 100 kb and other parts mention 1.1 Mb. Adding to the confusion is that the code (their Table S5) simulates 200 kb regions but employs parameters calculated for 1.1 Mb regions. 

**Bypassing Training Data by Importing Random Annotations.** Since Schrider and Kern [4] lacked a true training dataset based on actual genomic data with factual annotation, they simulated their own dataset and generated their own annotation for the simulated dataset. 

Innovatively, the authors devised the following quick fix to this seemingly intractable problem. They randomly selected 1.1 Mb regions from the human genome and used public datasets, such as phastCons, to annotate them. In the next stage, they generated a random sequence and copied the annotation of the real sequence onto the simulated one. For example, if the authors selected the region chr1:55000000–56100000 and if there was a conserved phastCons element at chr1:55000000–55000009, then the first ten bp of the simulated region would be marked as negatively selected (i.e., evolving in a neutral fashion). At no time were actual nucleotides from the human genome considered. Figure 2 illustrates, in an exaggerated fashion, the ridiculousness of this method.

**Classifying the Simulated Sequences into Five Classes.** Schrider and Kern [4] applied their SML classifier to the simulated data so that their classifier will assign the sequences to one of the five classes (experienced a hard selective sweep, linked to a hard selective sweep, experienced a soft selective sweep, linked to a soft selective sweep, and evolved neutrally, which presumably includes regions evolving under strict neutrality as well as sequences affected by purifying selection). 

The authors are very unclear about how the classifier’s decisions were made. For instance, they write: “For our classifications we simply took the class that S/HIC’s classifier inferred to be the most likely one, but we also used S/HIC’s posterior class membership probability estimates in order to experiment with different confidence thresholds”. We interpret this statement to mean that irrespective of the class inferred by the S/HIC classifier, the authors chose any threshold they fancied to make the final class determination. 

Schrider and Kern [4] argued that soft sweeps are the dominant mode of adaptation in humans. Analyzing the six human populations, they identified 1927 distinct selective sweeps patterns, 1776 (92.2%) of which were classified as being a soft sweep using their S/HIC method. An examination of Table S2 in Schrider and Kern [4] shows that the classification is biased towards “soft sweep” irrespective of the cutoff, which varies from 0.2 to 0.9 per population. In all those cases, S/HIC “classified” 72.22–99.15% of the segments as “softly selected” with a median of 93.04%. No doubt, this resulted in 73.1% of these sweeps being deemed “novel” as they do not exist in dbPSHP. To get these newsworthy results, it was essential for the authors to preselect populations that are included in dbPSHP, regardless of their heterogeneity. The “classifications” to “hard sweep” ranged from 0.85–27.78%, with a median of 6.96%. These are remarkable figures because hard sweeps are extremely rare in human populations (e.g., [39]). We also note that the exact balance between hard and soft sweeps depends on the yet undetermined distribution of mutational target sizes [40]. We also note that on theoretical grounds [20,21], the number of soft selective sweeps is expected to be much lower than that of hard selective sweeps.

There are other problems with the S/HIC classifications, one of which is that “adjacent windows are especially likely to receive identical annotations [3]. To overcome this difficulty, a secret (i.e., unpublished) “permutation algorithm” was implemented that considered the lengths of a run of consecutive windows assigned to each class per population. The run-length distribution was then obtained from the simulated data. By the end of this procedure, the windows are thoroughly reshuffled, which addresses the concern that adjacent windows have similar annotation but not the concern that certain features and classifications are inflated due to the over-classification in the adjacent windows in the first place. 

To summarize, Schrider and Kern’s classification of genomic regions into classes required not only an SML classifier but also a human intervention in the form of setting subjective thresholds that ranged widely in value. These thresholds, in turn, generated preconceived results that lack any scientific merit.

**Identifying Genetic-Element Enrichment in Selective Sweeps.** To complete their study that rests entirely on imaginary sequences onto which random annotations were thrust upon, Schrider and Kern [4] ask which “biological pathways” show “a strong enrichment” in genomic regions that were deemed to have been subject to selective sweeps. Here, it was expected that the authors would test the enrichment of actual biological features, such as certain metabolic pathways, certain gene families, or genes expressed in certain tissues. Alas, this is not the case. With the exception of the vague category “coding sequences”, which includes open reading frames for which we have no evidence of translation, let alone function, all other features that were found to be enriched in one class or another have nothing to do with real data. For example, the transcription factor binding sites were taken from the ENCODE project [41], which, as we all know, cannot be used to identify biological functions [42]. Another so-called functional dataset is COSMIC, which is a set of somatic mutations that have been observed in cancer cells [43]. Admittedly, a minuscule minority of these mutations may play some role in tumor suppression or progression; however, the vast majority of mutations in cancer cells are incidental and have neither a causative role in cancer nor in the progression of the disease. The closest Schrider and Kern [4] came to using real biological functions were the inferred functional categories from Gene Ontology [44], with a strong emphasis on “inferred” rather than “known”.

One such example of functional futility involves Schrider and Kern’s [4] “dramatic enrichment” of sweeps in genes that encode proteins that interact with one another. As far as “gene networks” are concerned, however, the term “interaction” is defined very broadly. To illustrate the problematics of using interactions as a validation method, we performed the following experiment. We selected 100 random protein-coding genes that had a HUGO (Human Genome Organization)/Gene Nomenclature Committee (HGNC) symbols [45] and used GeneMania [46] to identify genetic interactions among those genes. Of the hundred random sequences, only 17 had no “genetic interaction” with another gene. The remaining genes exhibited “genetic interactions” and “physical interactions” although none of these “interacting pairs shared a single biochemical pathway. By showing that random genes in our negative control exhibit extensive “genetic interactions”, we have demonstrated that Schrider and Kern’s [4] “interacting gene networks” is a meaningless concept and that the “dramatic enrichment” is biologically insignificant. Here, it may be worth mentioning Schrider and Kern’s [4] statistical choices. First, in all their calculations, they used one-tailed statistical tests, which are more likely to reject null hypotheses than two-tailed tests. Second, they did not employ any type of control. 

Finally, we would like to mention a common failure to most data-driven studies—lack of any follow-up. Schrider and Kern’s [4] study contains some results that *prima facie* seem remarkable. For example, of the 19 annotated features, the CEU population (Utah residents with Northern and Western European Ancestry) showed no significant enrichment for any annotation except “enhancers lost in humans since splitting with rhesus.” By contrast, all the other populations showed significant enrichment of more than 50% of the annotated features (Table S3 in [4]). The authors offer neither an explanation for this difference nor a reason why such a result is reasonable. Similarly, ClinVar pathogenic SNPs were significantly enriched only in PEL (Peruvians from Lima, Peru) despite the fact that this population has fewer ClinVar pathogenic mutations than YRI, CEU, and CHB (Figure S14 in [47]). As in all other cases, the authors are mum on these findings.

## 7. Discussion

Jesnsen [1] and Harris et al. [2] challenged the recent *“unfounded enthusiasm for soft selective sweeps in the evolutionary literature.”* In particular, Jensen [1] noted that while hard selective sweeps are well defined, and the genetic patterns expected under the hard sweep model have been precisely described in the literature, soft sweeps constitute a mixture of models with vastly different expectations. Harris et al. [2] carried out a detailed evaluation of the soft sweep results reported by Schrider and Kern [4]. They first noted that S/HIC results, as reported by its authors [3], already point to it being a poor predictor of sweeps. Even for a hard sweep with extremely strong selection (5000 < 2*Ns* < 50,000), the correct model is identified in fewer than 5% of simulated replicates, while hard sweeps were incorrectly classified as soft sweeps in nearly 50% of the replicates, as linked soft sweeps in 16% of the replicates, and as neutrally evolved sequences in 31% of replicates. In other words, under their own estimated demographic history, Schrider and Kern’s [3] S/HIC was unable to detect hard sweeps and has an extremely high false-positive rate in the direction of misclassifying them as soft sweeps when the demographic model is misspecified. Furthermore, Harris et al. [2] found that Schrider and Kern’s [3] presentation of S/HIC performances is purposely deceitful to give the impression that S/HIC performs well, when in reality, “misclassification is almost universally in the direction of falsely identifying soft selective sweeps,” the majority of which can be explained as misclassification of neutrally evolving regions.

The distribution of fitness also affects S/HIC’s predictions. When the selection in the training set is stronger than in reality, S/HIC overwhelmingly classifies hard sweep windows as soft. When the range of selection (2Ns ~ U[10, 1000]) is weaker than in Schrider and Kern (2Ns ~ U[166, 3333]), 47% of hard sweeps are misclassified as soft, and 46% of hard-linked sweeps are misclassified as soft-linked [2]. The choice of high selection coefficients that are uniformly distributed between 0.005 and 0.1 along the genome (to ensure that S/HIC’s performances would be optimal according to the simulations) is unrealistic and contributes to inflation of reported soft sweeps.

Here, we have shown that SML is inherently inapplicable to exploratory studies on evolutionary questions related to the driving forces of evolution for the simple reason that true empirical training datasets do not exist. As far as the study by Schrider and Kern [4] is concerned, the inappropriate methodology is further compounded by cherry-picking population data, logical missteps, internal inconsistencies, and the corruption of statistical practices.

Despite claims to the contrary, Schrider and Kern [4] have not provided *“an excellent example of how cutting-edge methods from computer science and statistics can be successfully brought to bear on long-standing questions in evolutionary biology”* [16]. What they did provide is yet another example of the inferiority of data-driven “science” relative to hypothesis-driven science. Allen [48] predicted that induction and data-mining, uninformed by ideas, can produce neither knowledge nor understanding. Interestingly, this prediction is a Popperian prediction that can be disproved by a single demonstration that a hypothesis-free process when applied to data, is sufficient to produce a gain in understanding. Schrider and Kern’s [4] study constitutes no such demonstration.

Our findings are similar to those of Harris et al. [2], who warned us against *“the growing tendency of invoking parameter-heavy, assumption-laden models of pervasive positive selection, and neglecting best practices regarding the construction of proper demographic null models.”* The purpose of Schrider and Kern’s “machine learning” is to create the appearance of sophistication, of doing something extraordinarily clever and original, so that anyone who challenges their work can be cowed into submission or be labeled stupid. Fortunately, the authors of this note are familiar with the parable in The Emperor’s New Clothes.

## Figures and Tables

**Figure 1 genes-12-00527-f001:**
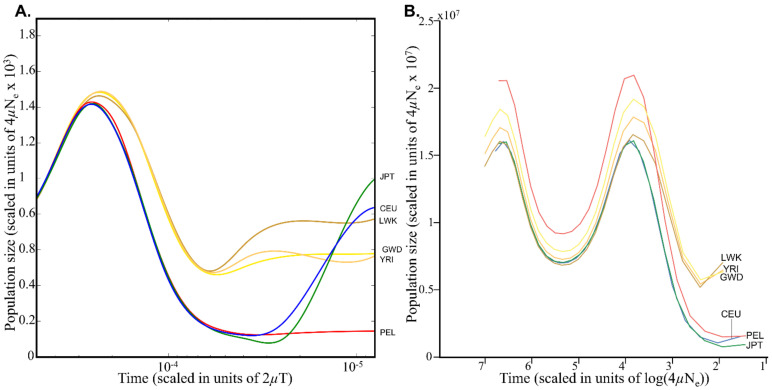
Population structure and demography. Changes in population size over time for six populations. (**A**) Changes to the effective population sizes over time in six populations inferred by PSMC. Lines represent the within-population median PSMC estimate, smoothed by fitting a cubic spline passing through bin midpoints. The original figure was published by Auton et al. (The 1000 Genomes Project Consortium 2015, Figure 2). This figure was created using code and data provided by Dr. Adam Auton to include only the relevant populations. The plot is log-scaled for the *X*-axis. (**B**) Plotting the Schrider and Kern’s [4] data for the Auton et al.’s figure. Schrider and Kern (2017) sampled 26 data points from (**A**) and scaled them by *θ* and *N_0_*. We *θ*-scaled the *X*-axis (to get each population on the same timescale) and log-scaled it to increase the similarity with (**A**).

**Figure 2 genes-12-00527-f002:**
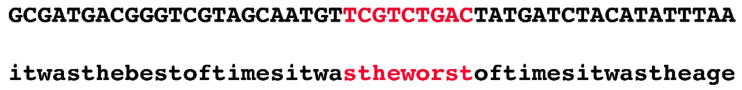
Illustration of the annotation method for the simulated “training data” used by Schrider and Kern [4]. Since Schrider and Kern [4] lacked true training data derived from a sample of the true genomic data with the features of interest, they simulated their own dataset. To annotate it so that it can be used to train their classifier, they randomly selected 1.1 Mb regions from the human genome, annotated them using public datasets like *phastCons*, and copied the annotation to their simulated data. To illustrate the problem with this approach, we start with a real sequence from the human genome (top) for which an annotation exists in phastCons. Let us assume that within this sequence, one region was found to be extremely conserved (red), i.e., subject to strong purifying selection. We then take another string of letters of identical length (bottom), call it the training sequence, and annotate the corresponding positions as “evolving neutrally.” If the “training” sequence is the start of the first sentence in *A Tale of Two Cities* by Charles Dickens (1859), then the string “… s the worst…” will be deemed to have evolved under purifying selection.

## Data Availability

No data were generated in this study.

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
