# Peer review of "On the Unfounded Enthusiasm for Soft Selective Sweeps III: The Supervised Machine Learning Algorithm That Isn’t"

_genes, 2021, doi:10.3390/genes12040527_

Round 1
Reviewer 1 Report
see attached pdf.

Author Response
See the enclosed file

Reviewer 2 Report
This Perspective piece by Elhaik and Graur addresses the growing trend of applying methodologies from the field of supervised machine learning to (SML) evolutionary genetics. The manuscript focuses on Schrider and Kern (2017), which used a supervised machine learning approach to find hard and soft selective sweeps in six human populations.
These paper has numerous major flaws, which I describe below before listing minor comments at the bottom of the review:
1) The most fundamental flaw of this paper is that it is based on a false premise: that evolutionary inference cannot work when it is based on simulation. The authors are right to point out that SML applications in evolution based on simulated "training" data are different from the traditional form of SML, but there is no reason why this should be a problem. Indeed, this is conceptually identical to approximate Bayesian computation (ABC), in which multivariate summaries of real data are compared to those of a large amount of simulated data. This approach has been around for over two decades (doi: 10.1093/genetics/145.2.505) and has proved useful for numerous types of evolutionary inference, including various model selection problems, and has become a mainstay for demographic inference. Indeed, model selection based on rejection-sampling ABC is extremely similar to k-nearest neighbor classification, an early SML algorithm, and one popular ABC model selection tool is essentially a random forest classifier (doi: 10.1093/bioinformatics/btv684). It is also worth noting that the ATLAS Collaboration is currently using simulations combined with SML to search for new particles (https://atlas.cern/updates/briefing/search-new-particles-machine-learning). Moreover, simulations are also often used by non-SML population genetic inference software such as LDhat and SweepFinder.
Thus, if the authors wish to show that SML is inherently inapplicable to evolutionary research, they must show that all simulation-based inference is so, rather than focusing on a single study. "Proof-by-example" does not suffice. On this note, the authors conclude their abstract, after listing some of their specific concerns with Schrider and Kern (2017), that those concerns lead them "to conclude that all evolutionary inference derived from so-called SML algorithms ... should be taken with a huge shovel of salt." This is a rather large and confusing leap that seems to defy logic.
2) The authors seem to take issue with the fact that simulation-based inference requires researchers to draw evolutionary parameters from prior distributions, which are often (but not always) uniform distributions designed to encompass a region of parameter space that the researcher hopes will include the true value. This is not at all unique to ABC and SML applications to evolution. Any software for constrained optimization of evolutionary parameters (e.g. dadi, the powerful tool for demographic inference tool which has been successfully applied to a variety of systems) will rely on user-defined prior bounds on these parameters. Thus, it seems that if the authors wish to demonstrate that SML cannot be used for evolutionary inference they will have to show the same for this type of method as well.
It is possible the authors disagree with the paradigm of performing model selection at all in population genetics, and would instead prefer that all analyses be based on a predefined number/fraction of statistical outliers (although of course this can't be done for things like demographic inference or recombination rate estimation). In this case, a refocused perspective piece weighing the pros and cons of model selection and outlier approaches could be beneficial to readers, although this has already been done effectively by others (e.g. chapter 10 of Hahn 2018; ISBN: 9780878939657).
3) While the authors do not present any coherent arguments for why evolutionary studies based on simulation or prior bounds around a multidimensional parameter space are invalid, they do present a number of criticisms of Schrider and Kern (2017), including their selection of study populations, simulation parameters, and downstream statistical analyses. Most of these criticisms are clearly false, and I will list some of the errors in the minor comments section below. However, the more important point is that none of these criticisms have anything to do with the applicability of SML to evolutionary analysis.
4) The manuscript contains numerous rhetorical flourishes that are unhelpful (e.g. Fig 2, the entirety of the section beginning on line 76), especially in light of the manuscript's numerous logical and analytical errors. These should probably be removed.
minor comments:
Line 27: discoal is a simulator, not an SML algorithm.
Line 61: The authors assert that evolutionary researchers have become interested in SML algorithms because of their success in e-commerce. Upon what have they based this assertion?
Line 195: The authors describe Schrider and Kern's range of selection coefficients as being skewed toward weak selection. I do not know any population geneticist who would refer to a mean value of s>0.05 as being low.
Lines 198-203: The language here, and also in the title of the section, seems to imply that Schrider and Kern have intentionally manipulated the parameters of their study to get a desired answer. I would suggest rephrasing or removing this baseless accusation.
Lines 247-249: It is not "remarkable" at all to point out that estimated demographic models will not correspond to the true population history. Due to imperfect optimization routines, identifiability issues, and unmodeled factors such as linked selection (e.g. doi: 10.1111/mec.13390), even demographic models that provide a good fit to the data will often be incorrect. The authors seem to be distorting the statement of fact as evidence of something nefarious.
Line 305: Simulating under a variety of neutral population mutation rates may be beneficial in systems where some regions will have much more polymorphism than others, for example due to variation in gene density.
Line 339: Population genetic simulators typically emit data only at segregating sites, so these numbers seem completely reasonable.
Line 358: I do not see the connection between phastCons elements and models of positive selection.
Line 388: "soft selection" and "soft sweeps" are two very different concepts.
Line 391: Another unfounded accusation that Schrider and Kern did something untoward in hopes of generating "newsworthy results".
Figure 1: The plot on the right of this figure is clearly incorrect. A cursory examination of the values of theta from Table S5 from Schrider and Kern (which shows the simulation command lines) reveals that the rank order of present-day population sizes from Schrider and Kern's simulations clearly matches that of Auton et al. This could imply that the authors of the present manuscript have omitted the present-day time from their plot. It is also worth noting that the plot from Auton et al. appears to be truncated on the left, so readers cannot tell from this figure if there is any difference between the two models.
Line 434: Regarding multiple testing, a quick search through Schrider and Kern (2017) reveals: "Because we tested each of S/HIC’s five classes for enrichment of a fairly large number of genomic features (supplementary table S3, Supplementary Material online), we corrected for multiple testing using false discovery rate q values following Storey (2002)."
Author Response
See the enclosed file

Reviewer 3 Report
This manuscript is to discuss on the limit of applying supervised machine learning (SML) to evolutionary studies. SML needs training data set which is supposed to be “true”. The authors discuss that there is no training datasets that are known to be a priori “true” therefore SML cannot used for evolutionary studies. As a case study, the authors discuss on an article that uses SML on human genomes to show that soft selective sweep is the dominant evolutionary force (Schrider and Kern 2017). The authors questions mainly two points for the work of Schrider and Kern (2017). One is about the training data set and about the simulation to generate it. Second is the choice of population data for analysis. Furthermore, the authors carefully examined the method of Schrider and Kern (2017) and found many problems.
Overall, I agree with the authors about SML on evolutionary studies and with that there are some problems on the work by Schrider and Kern (2017).
Although I can understand the authors’ frustrations at a paper by Schrider and Kern (2017), I feel that the tone of the manuscript is a little bit too strong and I am afraid that this tone can be too offensive as an scientific paper. Then I suggest the authors to reconsider the choice of words such as…
Line 24
“would put to shame most modern evangelical exegeses of the Bible.”
Line 66
“Such studies have been ballyhooed as …”
Line 362,
“Innovatively, the authors devised … “
Line 371
“the ridiculousness of this method.”
and so on.
I think the first paragraph of Discussion part starting from line 467 is not relevant to the current topic and I suggest the authors to remove this paragraph.
Author Response
See the enclosed file

Round 2
Reviewer 2 Report
Unfortunately, the authors have made no effort to address my concerns with their manuscript, and they have not constructed any coherent arguments in support of their reasoning, which remains fatally flawed. Indeed, their response displays an alarming degree of cognitive dissonance. I note a few instances and other major concerns below:
1) Most importantly, the authors state that articles describing ABC methodology have not used "a training dataset that was invented out of whole cloth." But these are methods that--EXACTLY like SML--require simulated training datasets constructed by their users. Don't just consider the papers introducing the methods, but also the hundreds of papers that have used ABC approaches when applied to their datasets. The prior distributions of evolutionary parameters used in these papers did not come from divine revelation. If it is inherently unacceptable to simulate training data to use a random forest to detect selective sweeps then it is unacceptable to use one to perform demographic model selection. If the authors disagree with this argument, which they seem to, then apparently it is acceptable to use a random forest to perform evolutionary inference in some cases, and the authors' central claim (SML cannot be used for evolutionary inference) is contradicted. If not, then my original comment stands and the authors must make a compelling argument as to why ALL applications of SML to evolution (demographic inference, recombination rate inference, recombination hotspot detection, detecting gene flow and adaptive introgression, inferring gene trees) are bound to fail. There are multiple papers that have worked on each of these problems and shown successful results. The authors must either put in the work of examining these other applications or abandon their claim.
2) In their responses the authors state that they cannot prove a negative, but in that same response they essentially assert that they have done such a proof by example! They state that it is"unlikely that other authors will be more successful with SML methods based on this demonstration." There is simply no support in their paper for such a conclusion! The authors state that they are only attacking Schrider and Kern's study, and that the perceived deficiencies of this study somehow proves that SML cannot be used for evolutionary inference, with the exception of the many studies that have apparently used SML in a way that, for reasons unclear, the authors do not find offensive (see above).
3) Skepticism is a virtue in science, but this paper ventures far beyond skepticism and into cynicism. The paper contains baseless accusations about the motivations of Schrider and Kern which these authors have refused to either substantiate (which of course they can't) or remove. On a related note, in their response to one of my comments the authors' brush away a relevant citation with an ad hominem attack (despite the fact that I shared this reference because I felt that it contained some arguments that the authors seem to agree with).The degree of cynicism on display here is disturbing, as it suggests that some in our community may be more concerned with whom they are attacking rather than the content of their arguments.
4) I will again point out a glaring error in the manuscript: If one looks at the command lines in Table S5 in Schrider and Kern, the values of theta clearly demonstrate that the rank order of the present-day effective population sizes among the six demographic models seem to match up quite nicely with those of Auton et al., whereas the authors' plot suggests otherwise. The authors have clearly made an error here despite their assertion to the contrary. That the authors would not carefully examine the validity of their own work even when specific errors have been pointed out raises serious concerns about their competence and rigor in evaluating the efficacy of any work that utilizes population genetic simulations.
5) This is further supported by the authors' response regarding the number of segregating sites emitted by the discoal simulator, which is simply "OK. This is a report of what this tool does." No, this is not okay. The authors are misrepresenting the information given by the simulator. Not every genomic region will have the same number of segregating sites--indeed this can vary dramatically due to sequence conservation, mutation rate variation, masking for data quality and repetitive elements, etc. Therefore any prudent simulation strategy will include such variation as well. This is also related to my original comment just above that one regarding variation in 4Neu, the relevance of which the authors clearly did not understand. 4Neu is a composite parameter and not all variation in it should be interpreted as variation in Ne. The authors should consider variation in u, the neutral mutation rate, which encapsulates all of the issues listed above.
All in all, this paper and the responses to the reviewers demonstrate a dangerous lack of understanding of how population genetic inference is carried out and as I noted in my original review the paper continues to be based on reasoning that is fatally flawed. It is unlikely to have much of an impact on established researchers who will see right through the authors' vacuous arguments, but it nonetheless poses a danger in that it would generate confusion among students entering the field both about the soundness of SML/ABC and also the ethics of attacking the character of researchers that one happens to disagree with.
Author Response
See in the enclosed file.
